# *M. mazei* glutamine synthetase and glutamine synthetase-GlnK1 structures reveal enzyme regulation by oligomer modulation

Maria A. Schumacher [1] ✉, Raul Salinas[1], Brady A. Travis[1], Rajiv Ranjan Singh [1] & Nicholas Lent[1]

Glutamine synthetases (GS) play central roles in cellular nitrogen assimilation. Although GS active-site formation requires the oligomerization of just two GS subunits, all GS form large, multi-oligomeric machines. Here we describe a structural dissection of the archaeal *Methanosarcina mazei* (*Mm*) GS and its regulation. We show that *Mm* GS forms unstable dodecamers. Strikingly, we show this *Mm* GS oligomerization property is leveraged for a unique mode of regulation whereby labile *Mm* GS hexamers are stabilized by binding the nitrogen regulatory protein, GlnK1. Our GS-GlnK1 structure shows that GlnK1 functions as molecular glue to affix GS hexamers together, stabilizing formation of GS active-sites. These data, therefore, reveal the structural basis for a unique form of enzyme regulation by oligomer modulation.

The enzyme, glutamine synthetase (GS; EC 6.3.1.2, L-glutamate: ammonia ligase) plays a central role in cellular nitrogen assimilation by incorporating ammonium into glutamine, which serves as a carbon and nitrogen donor for the biosynthesis of nucleotides, amino acids and lipids[1–3]. The GS reaction follows a two-step mechanism[1,2]. In the first step, GS binds glutamate and ATP and catalyzes phosphorylation of the glutamate γ-carboxyl group by ATP. In the second step ammonium is incorporated, leading to production of glutamine with the release of phosphate[1,2]. Underscoring its fundamental role in cellular physiology, GS is found in all organisms and phylogenetic studies point towards the genes encoding GS as being among the oldest described functional genes[4–6]. Based on sequence and structure, GS proteins have been grouped into three classes: GSI, GSII, and GSIII. Structures of GS proteins from the three classes have been obtained revealing that GSI and GSIII enzymes, which are found in bacteria and archaea, form dodecamers[7–10]. GSII enzymes are present in eukaryotes and while the *Phaseolus vulgaris* GSII was considered to be octameric, more recent structural data revealed a decameric oligomeric state for eukaryotic GSII[11–16]. The GSI enzymes have been further divided into subclasses GSI-α and GSI-β[1,2]. GSI-β and GSI-α enzymes are found in Gram-negative and low G + C Gram-positive (low G + C) bacteria,

respectively[1–4]. Despite the differences in the oligomerization and overall structure of the GS proteins from different classes, they all share a structurally conserved active site formed at the interface between two subunits in the oligomer[7–19].

Because GS proteins are critical for nitrogen metabolism, they are highly regulated. Studies have revealed diverse modes of GS regulation even within a given class of GS enzymes. In the GSI class, GSI-β enzymes are regulated by AMPylation (also termed adenylylation), while GSI-α enzymes are not AMPylated but are regulated by feedback inhibition by the product, glutamine[20–22]. Also distinct, some archaeal GS enzymes are regulated by 2-oxoglutarate[20]. Recent analyses have revealed another form of GS regulation involving interactions with other proteins[17–27]. For example, data have shown the symbiosome protein, Nodulin 26 (Nod26), interacts with soybean GS[23] and small proteins called IFs bind and inhibit the activity of cyanobacterial GS[24–26]. In Gram-positive bacteria, the transcription regulators TnrA and GlnR bind to GS and inhibit its catalytic activity[17,18,22]. Less is known about GS regulation in archaea. However, studies on the archaea, *Methanosarcina mazei* (*Mm*), indicated that *Mm* GS interacts with the small protein (sP26), consisting of 23 residues. In addition, *Mm* GS was shown to form a complex with the GlnK1 protein[27,28]. Notably, GlnK1 is a

[1]Department of Biochemistry, 307 Research Dr., Box 3711, Duke University Medical Center, Durham, NC 27710, USA. ✉e-mail: Maria.Schumacher@duke.edu

member of the PII protein family, which are pivotal regulators of nitrogen metabolism. Hence, the *Mm* GS-GlnK1 interaction links the key enzyme of nitrogen assimilation with a central integrator of nitrogen metabolism[29,30]. Initial data suggested the GS-GlnK1 interaction was inhibitory for GS, but subsequent analyses demonstrated it leads to *Mm* GS activation[27,28]. However, the molecular mechanism behind this protein-protein mediated regulation is unknown.

While long recognized as regulators of nitrogen metabolism, more recent data have shown that PII proteins coordinate and regulate multiple metabolic processes[29–35]. Two main classes of PII proteins have been identified, the GlnB and GlnK proteins[29–39]. These proteins both consist of 12–13 kDa subunits and have partially overlapping functions. Structures obtained for GlnB and GlnK proteins revealed that they harbor the same trimeric structure with each subunit containing a long, flexible loop, which has been called the T-loop due to the presence of a conserved tyrosine. This tyrosine is modified by uridylylation or adenylylation in some PII proteins, which impacts PII protein function[29,30]. GlnB and GlnK are also regulated by interactions with small molecules, including ATP, ADP and 2-oxoglutarate[29,30]. In *M. mazei*, there are two GlnK proteins, GlnK1 and GlnK2. However, only GlnK1 forms a complex with GS[27]. While 2-oxoglutarate binds both *Mm* GS and GlnK1, GS-GlnK1 complex formation requires neither ATP nor 2-oxoglutarate[27]. sP26 was found to bind both GS and GlnK1 but also was not required for the GS-GlnK1 interaction[27]. The GlnK1 protein is generated during nitrogen depletion, thus serving as a signal for the low nitrogen status of the cell[27]. Activation of *Mm* GS by GlnK1 binding drives an increase in cellular nitrogen incorporation.

Interestingly, while all known GlnK proteins are trimers, *Mm* GS is proposed to form a dodecamer. Hence, how *Mm* GlnK1 interacts with and regulates *Mm* GS is unclear. Here we describe biochemical experiments, X-ray crystallographic and Cryo-EM studies that elucidate the structural mechanism by which *Mm* GS functions and is regulated. Specifically, we report structures of *Mm* GS in its apo state, transition state and in complex with GlnK1. Strikingly, our cryo-EM structures of the apo *Mm* GS captured a heretofore unseen GS intermediate containing partially oligomerized hexamer rings. As the GS active sites are formed between GS subunits in the hexamer, this suggested oligomerization as a target for regulation. Indeed, our GS-GlnK1 structure revealed that GlnK1 trimers dock on each hexamer face of the GS double ring dodecamer to form a large GS-GlnK1 suprastructure in which GlnK1 subunits enforce *Mm* GS hexamers. Thus, the combined data unveil a unique mode of GS regulation involving protein-mediated hexamer modulation.

## Results

### Structure of apo *Mm* GS captures partially oligomerized GS
Although the *Mm* GS shares high sequence identities with GS proteins from *Staphylococcus aureus* (*Sa*), *Bacillus subtilis* (*Bs*), *Listeria monocytogenes* (*Lm*) and *Paenibacillus polymyxa* (*Pp*)[17,18] (55–59%) (Supplementary Fig. 1), previous studies suggested it displays different oligomeric properties compared to these enzymes[28]. To investigate this in more detail, we employed mass photometry (MP). MP utilizes low concentrations of sample that matches the physiological concentrations of many proteins. The method provides molecular weight analyses and thus information on oligomeric states. For these experiments we compared samples of apo *Mm* GS and apo *Bs* GS at concentrations of 75 nM. These experiments showed that 97% of the apo *Mm* GS was present as dimers with no dodecamers present. By sharp contrast, 68% of the apo *Bs* GS was distributed as a higher order oligomer consistent with a dodecamer with only 13% present as a lower molecular weight dimer species (Supplementary Fig. 2).

To next gain structural insight into *Mm* GS we utilized cryo-electron microscopy (Cryo-EM) (Methods). These experiments, performed with apo *Mm* GS at ~20 μM, revealed that 14% of the sample was captured in a partially oligomerized form with the remaining particles

adopting a dodecameric state (Fig. 1a–f, Fig. 2a, b; Supplementary Fig. 3a–f; Supplementary Table 1); particles corresponding to monomers/dimers were not processed due to their small sizes. The fully oligomerized *Mm* GS structure, which was resolved to 3.0 Å resolution, is composed of a dodecamer with stacked hexamer rings. Despite the relatively low resolution (6.9 Å), the partially oligomerized state of GS was evident in the structure and revealed that each ring contained only four subunits. The four subunits all make interactions with other subunits in the top ring. In addition, these subunits also all make inter-ring contacts with the four subunits of the second ring (Fig. 2a; Supplementary Fig. 3). Notably, although low resolution does not allow for structural details, the positions of the four rings overlap their positions within the fully oligomerized dodecamer (Supplementary Fig. 4).

As expected, structural homology searches showed that *Mm* GS subunits display the strongest similarity to subunits of Gram-positive bacterial GS structures, with root mean squared deviations (rmsds) of 0.5–1.0 Å for 430 similar Cα atoms compared to the *Bs* GS, *Lm* GS, *Sa* GS and *Pp* GS[18]. Similar to these GS structures, each *Mm* GS subunit is composed of 15 β-strands and 15 α-helices and each subunit is divided into a larger C-domain and an N-domain by helix α3 (Supplementary Fig. 1). Like the Gram-positive GS, the *Mm* GS active sites are formed at the interface between two subunits in the hexamer (Fig. 2a, b) and are comprised of five regions; the E flap (*Mm* residues 303–310), the Y loop (residues 369–377), the N loop (residues 235–247), the $Y^{179}$ loop (residues 152–161) and the D50´ loop (residues 56-71), which is the only active site region contributed from the adjacent subunit. The D50´ loop and E flap are arguably the most important active site regions as they contain the aspartic acid (D57 in *Mm* GS) that abstracts the proton from ammonium and the catalytic glutamic acid, Glu307, respectively. Like other GS dodecamers, the *Mm* GS dodecamer is formed by two interfaces; hexamer and inter-hexamer interfaces. The hexamer interfaces are located between subunits in each ring, and the inter-hexamer interfaces (also called thong interactions), are between subunits from each ring (Fig. 2a, b).

Our data show that, unlike the Gram-positive GS, *Mm* GS forms essentially no higher order oligomers or dodecamers at low concentrations (Fig. 2a, b). To gain insight into the different oligomer properties of these GS we compared their oligomer interfaces. The inter-hexamer/thong contacts in the *Mm* GS and the bacterial GS structures are essentially identical (Fig. 3a). By contrast, the hexamer interfaces differ in structure and sequence in a region corresponding to *Mm* GS residues 165–169 (Supplementary Fig. 1). In the bacterial GS, these residues are in a loop and the residue corresponding to *Mm* GS 167 is a leucine that inserts into a pocket of the adjacent subunit, contributing to the hexamer interface (Fig. 3b, c). In *Mm* GS residues 165-168 form an extra turn of helix and residues 167-168, which are an arginine and alanine, are positioned outside the oligomer interface; unlike the leucine, the arginine in *Mm* GS cannot bind within the hydrophobic pocket (Fig. 3c). Interestingly, *Mm* residues 165–169 are proximal to the catalytic $Y^{179}$ loop, suggesting this region could impact GS active site formation (Supplementary Fig. 1). Based on these analyses, we posited that residues 167–168 might play a role in the different oligomerization properties of these GS.

To test this hypothesis, we generated a *Mm* GS(R167L-A168G) mutant and performed MP experiments on the mutant protein. The MP data for the mutant, performed at 75 nM, showed that, unlike the WT Mm GS, it displayed higher order oligomer species. Specifically, while 45% of the sample was consistent with a dimer, 41% was present at a MW corresponding to 211 kDa and 12% was found at a MW consistent with a dodecamer (Supplementary Fig. 5). While the intracellular concentrations of *Mm* GS has not, to our knowledge, been reported, data indicate that in bacteria GS concentrations can, under some circumstances, reach μM levels[3]. Hence, to analyze the mutant and WT oligomeric states at higher concentrations we performed SEC experiments at 20 μM. Our previous SEC analyses on the *B. subtilis* GS

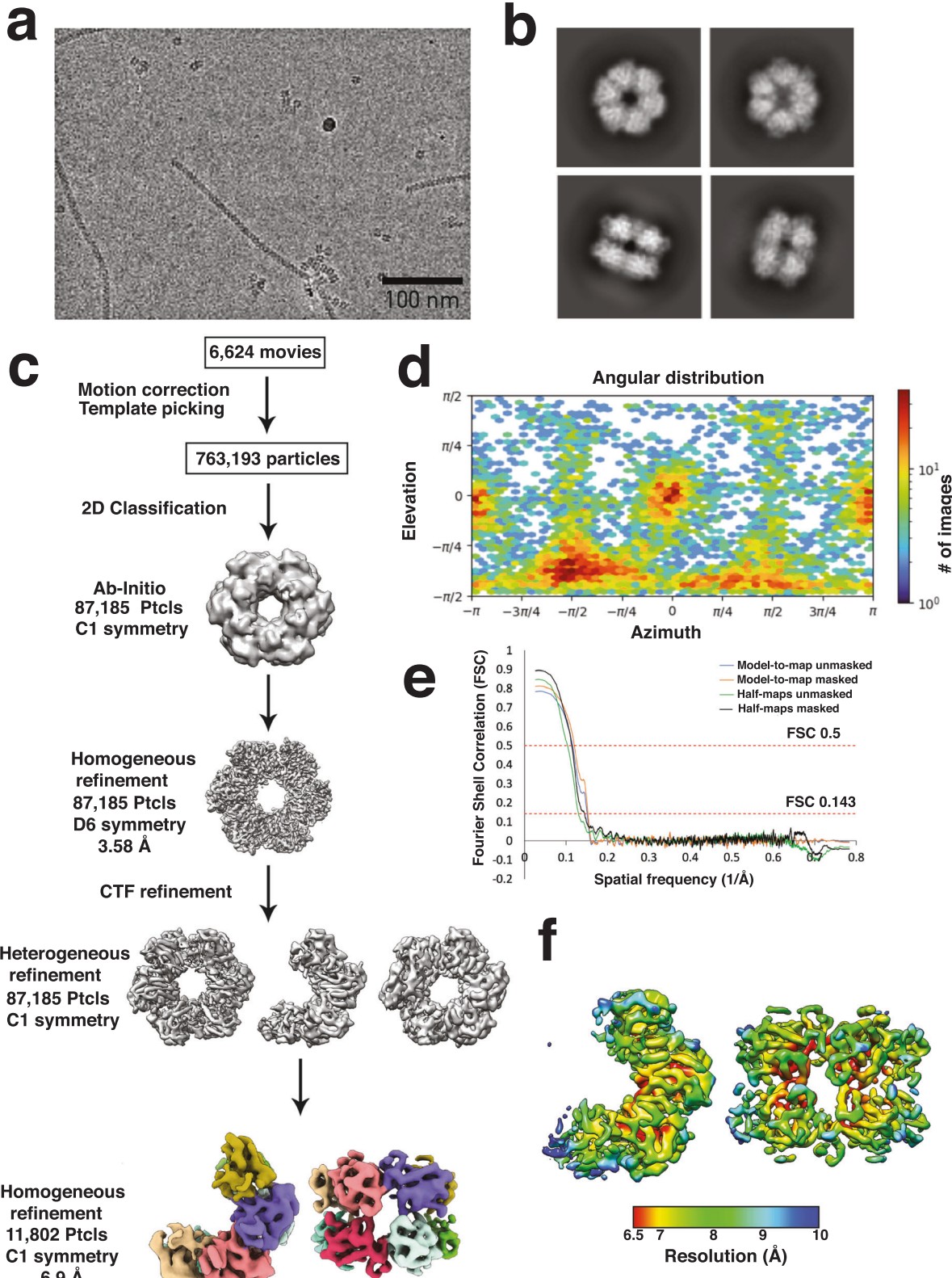

**Fig. 1 | Cryo-EM data processing and reconstruction of the partial *Mm* GS complex. a** A representative micrograph of the *Mm* GS complex on a holey gold grid. **b** A subset of the 2D classes showing top and side views of the complex. **c** Summary of the data processing workflow. **d** Angular distribution plot of the final particle set. **e** Masked and unmasked half-map and model-to-map FSC curves. **f** Final sharpened map colored by local resolution. For the dataset a total of 6624 movies were collected. The 3D reconstruction of the partial *Mm* GS structure shown in this figure was obtained from a total of 11,802 particles.

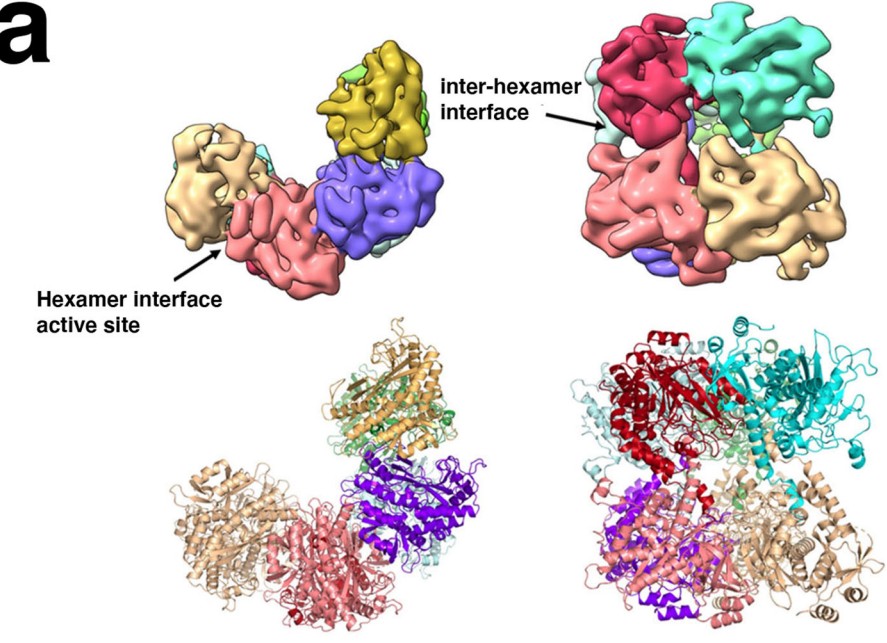

**apo *Mm* GS partial oligomer**

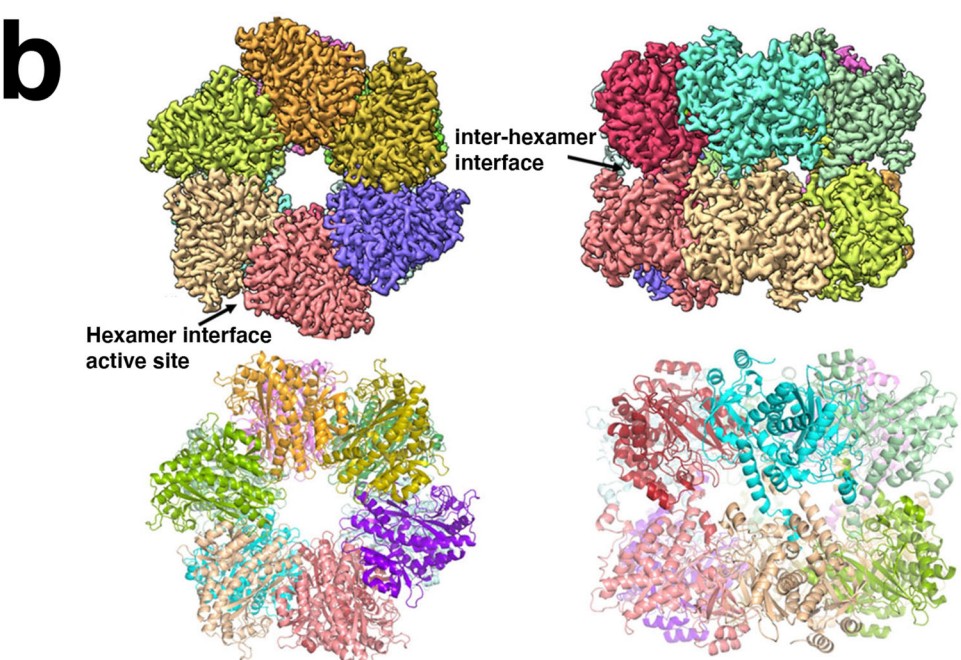

**apo *Mm* GS dodecamer**

**Fig. 2 | Apo *Mm* GS structures of dodecamer and partial oligomeric states.**
**a** Cryo-EM structure of the partial *Mm* GS structure, which contains only four subunits in each hexamer ring. Shown in top panel are two views of the cryo-EM map and below, the corresponding ribbon diagrams. The location of the interface between subunits in the hexamer (hexamer interface) and the inter-hexamer interface are labeled. **b** Structure of the apo *mM* GS dodecamer. Top panel shows two views of the cryo-EM map and below are the corresponding ribbon diagrams.

at similar concentrations revealed it eluted predominantly as a dodecamer[17]. In experiments with WT *Mm* GS and the mutant *Mm* GS, half of the WT GS sample eluted at a molecular weight consistent with a monomer/dimer and the other half, higher order oligomer. The *Mm* GS(R167L-A168G) mutant, by contrast, eluted with considerably more higher order oligomers (Supplementary Fig. 6).

These data revealed that the GS(R167L-A168G) mutant formed more higher order oligomers than the WT. However, the population of dodecamers in the GS(R167L-A168G) mutant was much less than we observed in the Gram-positive GS. These data were explained by a structure of the *Mm* GS(R167L-A168G) that we obtained at 2.45 Å resolution. The structure shows that the *Mm*

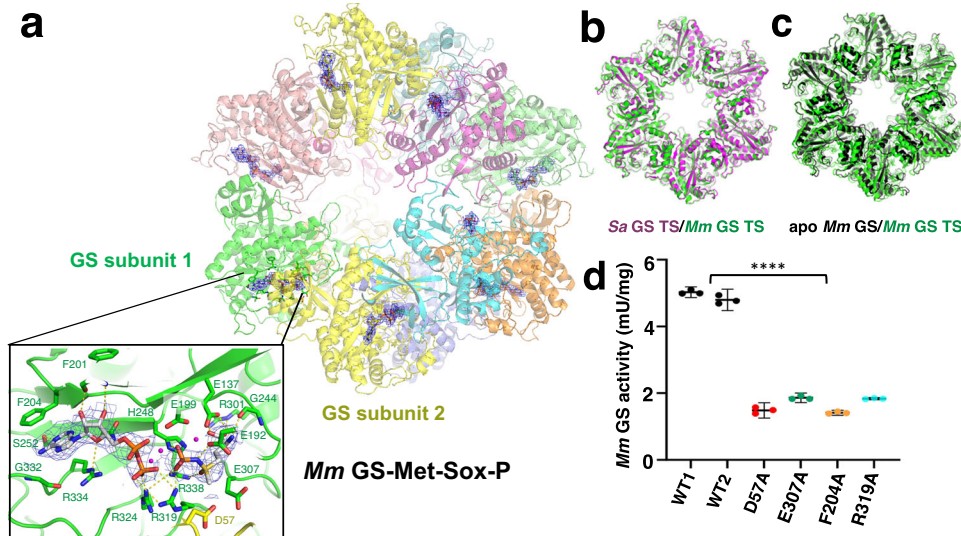

**Fig. 3 | Comparison of GS oligomer interfaces between Gram-positive and *Mm* GS. a** Overlay comparing the inter-hexamer/thong interface interactions of the *Mm* GS and S*a* GS showing they are essentially identical. The *Mm* subunits are colored green and light_green and the *Sa* subunits are pink and magenta and labels included to aid in distinction of the GS. The thong interactions are outlined to highlight their overall identical structure. **b** Superimposition of the *Mm* GS and S*a*

GS hexamer interface. The *Mm* GS and *Sa* subunits are colored as in Fig. 2a. Notice the different location of the loop in the *Sa* subunit, providing key subunit interactions with the neighboring subunit, not present in the *Mm* structure. **c** Close-up of the *Sa* GS loop interaction in Fig. 2b showing residues that differ between the two. Specifically, the RA residues in *Mm* GS are replaced by LG residues in *Sa* GS and hence cannot make the same hexamer interface interactions.

**Fig. 4 | Cryo-EM structure of *Mm* GS-Met-Sox-P ADP TS complex. a** Overall structure of the *Mm* GS TS complex with each subunit of the GS dodecamer colored a different color. The resultant omit $mF_o$-$DF_c$ map in which both the Met-Sox-P and ADP molecules were omitted during refinement is shown as a blue mesh (contoured at 3 σ) over the whole structure. Shown below the dodecamer is a close-up view of the active site formed between the green and yellow GS with residues making contacts to the Met-Sox-P and ADP shown. **b** Overlay of the *Mm* GS TS (green) with the *Sa* GS TS (magenta) showing they adopt the same overall structure

(rmsd of 0.8 Å for 430 Cα atoms). **c** Overlay of apo *Mm* GS dodecamer (black) onto the *Mm* GS TS structure (green) underscoring the large-scale structural changes that occur upon TS formation. **d** GS enzyme assay testing the effects of active site mutations on activity compared to the WT (WT1 and WT2). The results are the average of 3 measurements with error bar representing standard deviation (SD). Two-Way ANOVA using the software GraphPad Prism 9 was performed. The *P* value for all the Source of Variation (Interaction, Row Factor and Column Factor-as specified in the software) are statistically significant (<0.0001) in all cases.

GS(R167L-A168G) subunits are similar to the apo WT *Mm* GS (rmsd = 0.49 Å for 432 corresponding Cα atoms). Hence, the R167A and A168G substitutions did not result in these residues adopting the structure observed in the Gram-positive GS in which the residues inserted into the adjacent subunit (Supplementary Fig. 7a, b). However, the GS(R167L-A168G) structure shows that the substitutions led to the formation of a tighter interface compared to WT GS (Supplementary Fig. 7a, b). These analyses explain the intermediate impact on oligomerization observed for the GS(R167L-A168G) mutant compared to WT *Mm* GS and Gram-positive GS. These data thus support that residues 167-168 play a role in stabilization of the GS hexamer interfaces of these proteins, but indicate that the hexamer interface, not surprisingly, depends on more than just these residues.

### Structure of *Mm* GS transition state complex

Structural studies on GS have shown that formation of the transition state (TS) leads to large conformational changes, which include structural alterations in residues corresponding to 157-164 in *Mm* GS[18]. Because these residues are adjacent to the hexamer interface, which

adopts a different structure in *Mm* GS compared to the bacterial GS, it raised the question of whether *Mm* GS might employ a different catalytic mechanism and form a different TS. To address this question, we reacted *Mm* GS with the GS inhibitor L-methionine-S-sulfoximine (MSO) and ATP and obtained a 2.7 Å cryo-EM structure (Fig. 4a; Supplementary Fig. 8a–f) (Methods; Supplementary Table 1). MSO has been used to trap the TS in other GS as it is phosphorylated to Met-Sox-P and because the Met-Sox-P methyl group occupies the ammonium site, it functions as a TS mimic[1–3]. The resultant *Mm* GS TS complex has the same structure as has been observed in other GS TS (Supplementary Fig. 9; Fig. 4b), with the Met-Sox-P and ADP bound at the active site formed between GS subunits, supporting that *Mm* GS employs the same catalytic mechanism (Fig. 4a). Notably, the proximity of the region composed of residues 167-168, which form a distinct interface in the apo *Mm* GS structure compared to the low G + C GS, did not impact *Mm* TS formation; the *Mm* GS TS adopts the same active site and loop structure for residues 167-168 as the bacterial GS TS (Supplementary Fig. 10). Like other GS, *Mm* GS undergoes large conformational changes compared to its apo form upon Met-Sox-P formation (Fig. 4c).

In the *Mm* GS TS structure, the ADP is anchored into the active site pocket, next to the Met-Sox-P (Supplementary Fig. 10). The ADP adenine ring is contacted by GS residues Ser252, Phe204, Gly332 and Arg334, while the ADP β-phosphate is hydrogen bonded to Arg319 and Arg324 (Fig. 4a). The Met-Sox-P is contacted by GS residues His248, Arg319, Arg324 and Arg338. Because Arg319 and Arg324 interact with both ADP and the Met-Sox-P, they help orient the molecules in the active site. At the other end of the pocket, the side chain of *Mm* GS residue Arg301 contacts both oxygens of the Met-Sox-P carboxyl moiety and the sulfoxime atoms are bound by *Mm* GS residues Glu137, Glu192 and Glu199. GS catalytic residues Glu307 and Asp57 cover the active site, which positions the Glu307 side chain for proton abstraction of the ammonium (Fig. 4a). The interactions of GS residues from both subunits also stabilizes the GS oligomer, enabling catalysis. Although the *Mm* GS TS structure and interactions are similar to other GS, to further test the *Mm* GS TS structural model we mutated *Mm* GS residues Asp57, Phe204, Glu307 and Arg319, which the structure indicates are key for Met-Sox-P formation, to alanines and performed enzyme assays (Methods). Consistent with our structure, these mutations led to significantly impaired *Mm* GS activity (Fig. 4d).

### Structure of the *Mm* GS-GlnK1 complex

Recent studies demonstrated that *Mm* GS is positively regulated by forming a complex with the nitrogen regulatory protein, GlnK1[27]. To determine the molecular mechanism for this interaction and the basis for GlnK1-mediated GS activation, we created and purified the *Mm* GS-GlnK1 complex and obtained its crystal structure to 2.3 Å (Methods; Supplementary Table 2). Strikingly, the structure reveals the *Mm* GS dodecamer is sandwiched between two GlnK1 trimers (Fig. 5a). The α1 helices of *Mm* GS form a crown-like structure around a large central pocket at the top of each hexamer. The GlnK1 trimers slot into each of these hexamer pockets. The GlnK1 trimer core, which is structurally similar to other PII proteins, provides only long-range electrostatic interactions with GS; GlnK1 core residues Arg21 and Lys25 interact with GS residue Glu19 (Fig. 5b). Indeed, almost all the contacts to *Mm* GS are provided by residues from the GlnK1 T-loops, which extend from the trimer core and insert into the crevices formed between GS subunits.

T-loops are typically disordered in apo PII structures, but in the *Mm* GS-GlnK1 complex T-loop residues 44-52 form a helix that fits snugly into the GS crevices (Fig. 5a–c; Fig. 6a, b). Each T-loop makes extensive contacts with residues 20–23 and 87–92 from one GS subunit and residues 19–20, 84–87, 171–181 and 220-228 from a second GS subunit (Fig. 6a, b). Hydrogen bonds affix the T-loops into the GS pockets, including contacts from GS residues Arg20 and Glu19 to the carbonyl oxygens of GlnK1 residues Gly44 and Tyr45 and the GlnK1 Tyr45 side chain, respectively. GlnK1 residue Arg50 makes several contacts to GS, including a bidentate hydrogen bond to GS residue Asp21 as well as the carbonyl oxygens of Ala89 and Thr90. The GlnK1

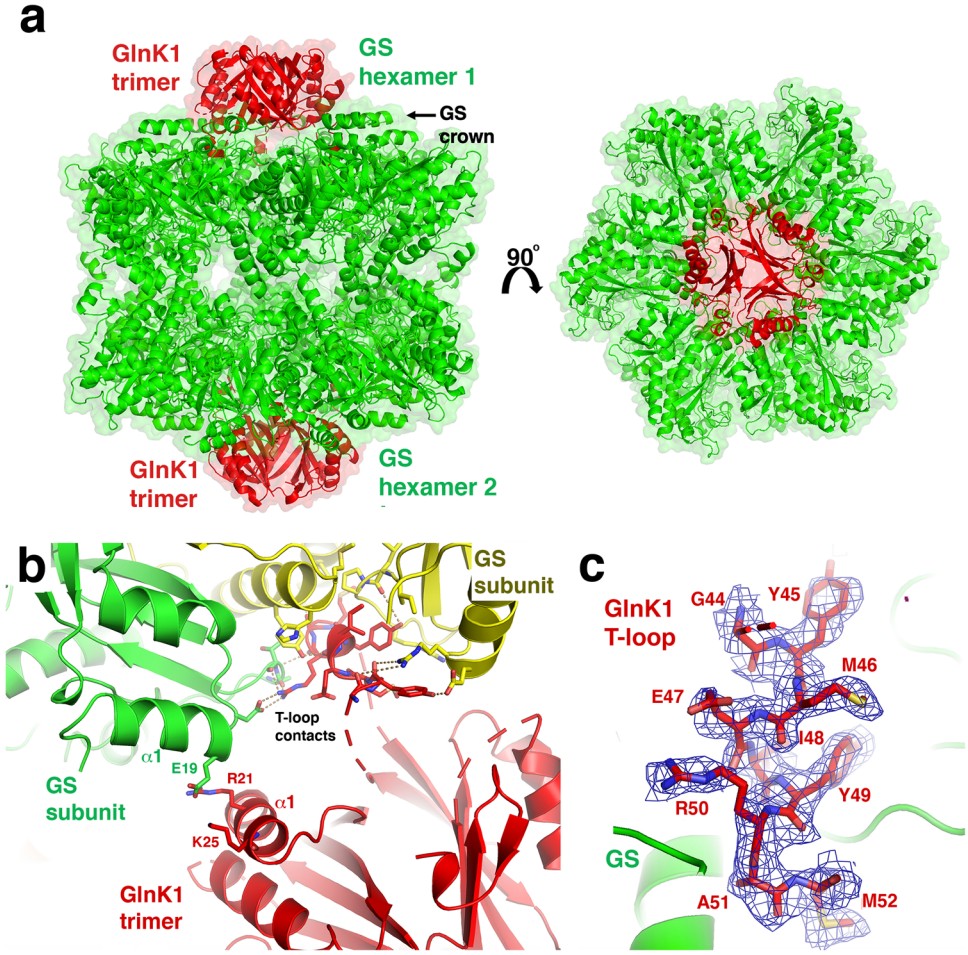

**Fig. 5 | Overall structure of the *Mm* GS-GlnK1 complex. a** Ribbon diagram of the *Mm* GS-GlnK1 complex with the GS dodecamer colored green and the two GlnK trimers colored red. Shown are two views of the complex related by a rotation of 90°. **b** Ribbon diagram of the GS-GlnK1 interaction zoomed out to show the electrostatic contacts between GlnK1 residues Arg21 and Lys25 and GS residue Glu19. The T-loop contacts are included to indicate their relative to the GlnK1 trimer body. **c** Close-up of the GlnK1 T-loop binding site in GS. T-loop residues are shown as sticks and colored red. The two interacting GS subunits are colored green and labeled. Also shown is the 2mF$_o$-DF$_c$ map contoured at 0.7 σ around the GS interacting GlnK1 T-loop residues. To avoid bias, the map shown is that calculated before addition of the GlnK1 residues including the T-loop.

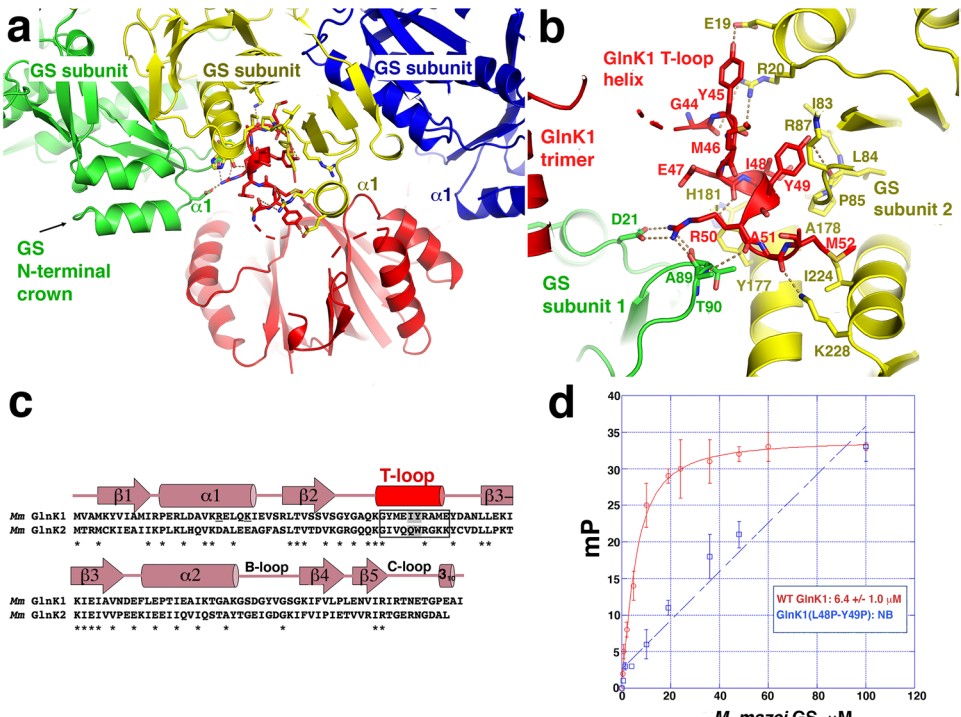

**Fig. 6 | *Mm* GS-GlnK1 T-loop contacts. a** Ribbon diagram showing GlnK1 T-loop interactions with two *Mm* GS subunits and location of the GS crown into which the GlnK1 trimer slots. Different GS subunits are colored, green, yellow and blue and GlnK1 is colored red. The helical region of the T-loop that docks between GS subunits is shown as sticks and the GS crown region formed by the GS N-terminal helices are labeled. **b** Ribbon diagram close-up of the T-loop binding pocket located between GS subunits. One GS subunit is colored green and the other yellow. Another non-interacting GS subunit is blue. GlnK1 is colored red. Residues that make interactions are shown as sticks and labeled. Hydrogen bonds between T-loop and GS residues are shown as yellow dotted lines. **c** Sequence alignment of *Mm* GlnK1 and *Mm* GlnK2. The location of the GlnK1 secondary structural elements, including the T-loop, are shown above the alignments, and labeled. Conserved residues between the proteins are indicated by asterisks below the alignment. Boxed is the region of the T-loop that contacts GS and the two residues that were mutated to test effects on GS binding are highlighted in grey. Underlined are the two positive residues that make electrostatic interactions with *Mm* GS. **d** FP binding isotherm showing the interactions of WT *Mm* GlnK1 and *Mm* GlnK1(I48P-Y49P) with WT *Mm* GS using F-ATP as a probe (Methods). WT GlnK1 bound WT GS with a $K_d$ of 6.4 μM ± 1.0 μM. *Mm* GlnK1(I48P-Y49P) showed nonsaturable binding (NB) to GS. The results are the average of three measurements with error bars representing SD.

Arg50 carbonyl oxygen also interacts with the amide nitrogen of GS residue Thr90. Finally, Lys228 from GS helps affix the T-loop helix within the GS binding site by contacting the GlnK1 Ala51 carbonyl oxygen (Fig. 6b).

In addition to hydrogen bonds, the GS-GlnK1 interaction includes hydrophobic and van der Waals contacts between GlnK1 T-loop residues Ile48 and Tyr49 and GS residues Ile83, Pro85, His181 and the side chain of Arg87. The interaction with His181 requires the rotation of its side chain and is the only interacting region of GS that undergoes significant structural changes upon GlnK1 binding. The GlnK1 Ala51 side chain makes van der Waals contacts with GS residues Tyr177 and Ala178 and the side chain of GlnK1 residue Met52 sits in a hydrophobic pocket formed by GS residue Ile224 and the side chain atoms of Lys228 (Fig. 6b). The finding from the GS-GlnK1 structure that the GlnK1 T-loops brace together GS hexamer subunits indicates oligomer stabilization as the GlnK1-mediated activation mechanism.

### GlnK1 is a helical adhesive for GS stabilization

To date, few structures have been obtained of PII-target protein complexes. Similar to the *Mm* GS-GlnK1 structure, most of these complexes revealed the T-loop as the PII interacting region[36–42]. In the GlnK-AmtB, PII-N-acetylglutamate kinase (NAGK) and PII-PipX complexes the T-loops form extended structures that interact with their partner protein[33–39]. The *Mm* GS-GlnK1 complex is distinct from these structures in that the GS interacting T-loop forms a helix. Thus, the *Mm* GS-GlnK1 complex expands the range of structures the T-loop is capable of adopting when interacting with partner proteins. As noted, the *Mm* GlnK2 protein is unable to bind GS[27]. The *Mm* GlnK1 and GlnK2

proteins share 62% similarity (42% identity). Despite the sequence homology, our GS-GlnK1 structure reveals that the residues in GlnK1 that contact *Mm* GS are not conserved in GlnK2. For example, GlnK1 residues Arg21 and Lys25, which make electrostatic interactions to GS are substituted with acidic residues in GlnK2 (Fig. 6c). Further, GlnK1 residues Tyr45, Ile48 and Tyr49, which make multiple contacts to GS, are substituted to isoleucine, glutamine and tryptophan in GlnK2, which cannot make the same interactions with GS (Fig. 6c). In addition, to bind GS, GlnK1 T-loop residues 44–52 must form a helix. Secondary structural analyses indicate that while the T-loop region of GlnK1 has high helical propensity, this region GlnK2 does not (Supplementary Fig. 11). Hence, these data indicate that substitution of key GS contacting T-loop residues in GlnK2 as well as a lack of helical propensity in the T-loop region of GlnK2 is why it does not interact with *Mm* GS.

However, to further probe our GS-GlnK1 structural model, we generated a GlnK1 mutant in which we substituted Ile48 and Tyr49 with prolines. These substitutions would be predicted to not only prevent contacts with GS but also prevent helix formation. To analyze binding, we developed a fluorescence polarization (FP)-based assay using Fluorescein-12-ATP (F-ATP), which contains a fluorescein attached to the ATP N7 atom (Methods). In this assay, GS is first titrated into a reaction cell containing 1 nM ATP N7 until binding is saturated. Next, the GlnK1, which cannot bind the F-ATP due to the location of the attached fluorescein tag (see Methods), was titrated into the reaction containing the ATP bound GS. GlnK1 binding to the GS was denoted by an increase in millipolarization and showed saturation when using the WT GnK1, with an apparent affinity of 6.4 ± 1.0 μM. The same experiment was performed using the GlnK1(L48P-Y49P) mutant and showed

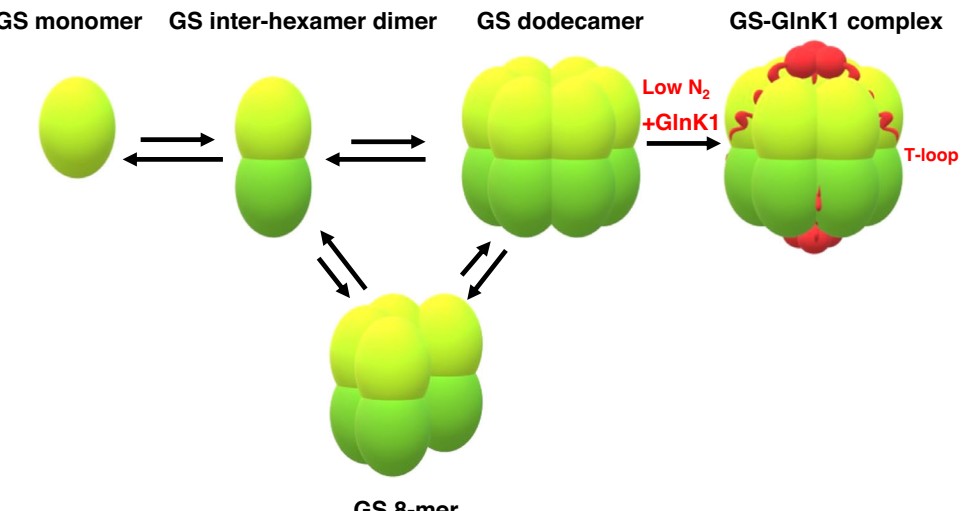

**Fig. 7 | Schematic showing *Mm* GS-GlnK1 regulatory circuitry.** The left equilibria indicates that *Mm* GS forms unstable dodecamers. Under conditions of low $N_2$, GlnK1 is expressed. GlnK1 forms a trimer that binds to the GS and stabilizes the dodecamer form of GS by inserting its flexible T-loops between GS subunits. The T-loops fold into helices that act as molecular glue to tie together the dodecamer. As the active sites are formed between adjacent subunits in the hexamer, this stabilization enhances GS activation. The figure was made using Microsoft paint (https://canvaspaint.org/#local:b9ce4db7f0fa8).

no saturable binding to GS (Fig. 6d). Thus, these data support the structural model.

## Discussion

GS enzymes are critical for nitrogen assimilation in all organisms. While different structural classes of GS enzymes have emerged, the active sites of these enzymes, which in all GS are formed between two subunits, have remained remarkably structurally and functionally conserved. Although GS enzymes require only two subunits to generate a functional active site, they have all evolved to form large multi-subunit oligomers. Why this is the case has represented an interesting enigma. Substantial information has been obtained regarding GS biology, structure and function in the last few decades allowing for a detailed understanding of the molecular mechanisms of these important enzymes. These studies show that the presence of a large GS oligomeric machine presents the opportunity for cooperativity[10]. But the data presented here reveals another role for oligomerization, which is as a route for regulatory input. Given its central metabolic role, the regulation of GS activity is vital for maintaining sufficient levels of nitrogen metabolites during changing nutritional conditions.

In particular, our studies revealed the detailed, molecular basis for regulation of a GS via modulation of its oligomer state (Fig. 7). Indeed, MP studies showed that apo *Mm* GS forms essentially no higher order oligomers at low concentrations. Using cryo-EM of *Mm* GS at 1 mg/mL, we captured a partially oligomerized form of *Mm* GS that retained the double-stacked oligomer contacts but lacked two subunits in each *Mm* GS ring. Previous studies had shown that when nitrogen levels drop in *Mm*, the GlnK1 protein is expressed and binds GS, enhancing its activity[27,28]. To elucidate the mechanism of GS activation by *Mm* GlnK1 we solved the high-resolution crystal structure of the complex. Aside from structures of Gram-positive GS enzymes in complex with C-terminal peptide regions of GlnR and TnrA, this represents the only structure of a GS in complex with a target protein. The GS-GlnR and GS-TnrA complexes showed that the C-terminal peptides both bound in the GS active site to favor the inactive site. Interestingly, in some Gram-positive GS, the interaction with GlnR leads to an oligomeric transition of the GS from a dodecamer to a tetradecamer[17,18]. However, this oligomeric change was not required for GlnR-mediated inhibition as not all the GS undergo this GlnR-driven transition such as *S. aureus* but still show enzymatic inhibition[18]. By contrast, our GS-GlnK1 structure revealed GlnK1 does not bind GS active site residues. Instead, by binding between subunits, it stabilizes the GS hexamers and hence the active enzyme state (Fig. 7).

Data from previous studies suggests that oligomer dynamics are observed in several GS. For example, studies showed that the GS enzymes from *Escherichia coli*, *Ruminococcus albus 8*, *Neurospora crassa*, and the plant *Phaseolus vulgaris* are present in solution as both lower molecular components and higher-order oligomers, indicating an oligomer equilibrium exists that could be utilized in regulation[40–42]. A recent study by Chen et al. showed evidence for regulation of the plant *Camellia sinensis* (*Cs*) GS involving oligomerization[19]. In this case, oligomer destabilization was noted within the inter-ring interfaces, which are not involved in active site formation. The data, however, suggested that these inter-ring interactions effect catalytic residues in the enzyme. Subsequent in vivo studies suggested that the 14-3-3 scaffold protein may interact with *Cs* GS to impact its oligomerization. Future work is needed to assess if the interaction is direct and determine impacts on *Cs* GS activity. Our work has revealed how binding of the central nitrogen regulator, GlnK1, stabilizes the key interface for GS activity; the hexamer interface. Notably, studies suggest a similar interaction may occur between GS and GlnK in other archaea, including *Haloferax mediterranei*[43]. Thus, these data indicate that the targeting of GS oligomerization may be a broadly employed mechanism for regulation of this key enzyme.

## Methods

### Protein purification

The genes encoding *Mm* GS, *Mm* GlnK1 and mutant forms of the proteins were purchased from Genscript Corporation and subcloned into pET15b such that an N-terminal His-tag was expressed on each protein for purification (Piscataway, NJ, USA; http://www.genscript.com). *Escherichia coli* C41(DE3) cells were transformed with these expression vectors. Cells with each expression construct were grown at 37 °C in LB medium with 0.10 mg/mL ampicillin to an $OD_{600}$ of 0.3-0.4, then induced with 0.50 mM isopropyl β-d-thiogalactopyranoside (IPTG) at 15 °C overnight. For each purification, 9 liters of bacterial culture was grown. Cells were harvested by centrifugation and then resuspended in 100 mL Buffer A (50 mM Tris HCl pH 7.5, 300 mM NaCl, 5% (v/v) glycerol, 5 mM $MgCl_2$, 0.5 mM β-mercaptoethanol (βME)). Also added was 100 μL protease inhibitor cocktail stock (which contains 100 μM

aprotinin, 1 mM leupeptin, 1 mM pepstatin A) Roche) and 10 μL of a solution of 100 mg/mL DNase I (Roche) to each 100 mL resuspended cells. The resuspended cells were then disrupted with either a sonicator or microfluidizer and cell debris was removed by centrifugation (18600 g, 4 °C, 60 min). For each protein, the supernatant was loaded onto a cobalt NTA column (TALON Superflow histidine-tagged protein purification resin). The column was generated by pouring the resin into a Bio-Rad glass column and equilibrating before protein addition with Buffer A overnight. After protein application, the column was washed with 300–500 mL of 2 mM imidazole in Buffer A and eluted in steps with 5 mM, 10 mM, 20 mM, 30 mM, 40 mM, 50 mM, 100 mM, 200 mM, 300 mM and 1 M imidazole in Buffer A. Fractions were analyzed by SDS-PAGE and those containing the protein were combined.

### Size exclusion chromatography (SEC) experiments
SEC studies were performed using a SUPERDEX™ 300 pg column and a ÄKTAprime plus. The buffer used for all the runs was 25 mM Tris-HCl pH 7.5, 150 mM NaCl, 1 mM MgCl$_2$, 5% (v/v) glycerol. Fractions were concentrated individually using Sigma-Millipore concentrators prior to column application. Samples were loaded using a 1 mL (final volume) syringe. The SEC studies were carried out on apo Mm GS and apo Mm GS(R167L-A168G) both at 1 mg/mL. The elution volumes of each sample were compared with a series of protein standards to determine the molecular weights. The standards used for calculation of the standard curve were from the Gel Filtration Markers kit (Sigma Millipore Cat# MWF200). The standards were cytochrome c (12.4 kDa), carbonic anhydrase (29.0 kDa), bovine serum albumin (66.0 kDa), yeast alcohol dehydrogenase (150.0 kDa), β-amylase from sweet potato (200 kDa) and Blue Dextran (2000 kDa).

### Glutamine Synthetase (GS) enzyme assay
To interrogate GS enzymatic activity, we utilized the Biovision colorimetric GS activity kit (Cat K2056-100). In this assay, the ADP generated from GS activity is utilized in a subsequent reaction in the presence of ADP converter, developer mix and ADP probe to generate a colorimetric product read at an absorbance of OD$_{570}$. For these assays, apo WT Mm GS and apo Mm GS mutants were buffer exchanged into the GS Assay Buffer from the kit. The protocol included with the kit was used for the assays and the absorbance was measured immediately at 570 nm using a Molecular Devices SpectraMax M5, after reaction initiation. In these experiments Mm GS proteins were present at 60 μM. The measurements were done in kinetic mode at room temperature (rt) at 5 min intervals. One unit of GS activity was defined as the amount of enzyme that produces 1 μmol of ADP per min at pH 7.2 at 37 °C. The sample sizes for WT and mutant GS were both 15 μL The samples were all concentrated to the same OD (1.0) before the reaction, in order to use a multichannel pipette to aliquot all the samples simultaneously. This allowed the reaction times to be the same during the incubation period, across all samples. Three independent replicates were performed for each sample and sample background. To analyze the data for significance in difference between WT and mutants, we performed Two-way ANOVA using the software GraphPad Prism 9.0.0. The P value for the Source of Variation (Column Factor- as specified in the software) is statistically significant (<0.0001) between WT and mutants. The error bars represent the Standard Error of the Mean. Details are provided in the source data file.

### Fluorescence polarization (FP) binding experiments
To measure Mm GlnK1 binding to Mm GS, a fluoresceinated ATP (Fluorescein-12-ATP; Perkin Elmer) was used as a signal. Fluorescein-12-ATP contains fluorescein attached via an extended linker to the A7 atom of ATP. The Mm GS structure revealed that the A7 atom of ATP is solvent exposed and would bind F-12-ATP. By contrast, while PII proteins such as GlnK1 also bind ATP, analyses of PII-ATP structures revealed that an ATP with a tag attached to the N7 atom would likely encounter steric clash. Hence, in these experiments, in the first step, the Mm GS was titrated into the sample cell containing 1 nM F-12-ATP until binding saturation was reached. Next, GlnK1 (WT or the GlnK1(I48P-Y49P) mutant) was titrated into the same cell. The resultant increase in mP indicated GlnK1 binding to GS. Notably, prior to this, experiments were done to show that there was weak to no binding of the F-ATP to GlnK1. For these experiments, the sample cell contains 1 nM of the F-ATP in a buffer of 25 mM Tris pH 7.5, 100 mM NaCl, 1 mM MgCl$_2$. The resultant data were plotted using KaleidaGraph Version 4.5 for Mac; serial # 8011073 (Synergy Software) and the curves fit to deduce binding affinities. Three technical repeats were performed for each curve.

### Mass photometry (MP) experiments
MP experiments was done using a Refeyn OneMP instrument (Oxford, UK). The experiments were performed using microscope coverslips, cleaned by sequential sonication in isopropanol (HPLC grade, and Milli-Q H2O (15 min each)), followed by drying with a clean pressurized airstream. Clean coverslips were assembled into the flow chamber, and silicone gaskets were positioned on the glass surface for sample loading to hold the sample drops with 4×4 wells prior to measurements. Contrast-to-mass calibration was achieved by measuring the contrast of proteins in the native marker protein standard mixture (NativeMark Protein Standard, Thermo Fisher Scientific) to generate a standard calibration curve at 100-fold dilution; 1048 kDa, 480 kDa, 146 kDa, and 66 kDa molecular masses were used to fit the calibration curve with an R$^2$ value of 0.999 and a Max mass error of 10.1% in the Refeyn DiscoverMP v2023 R2 software. The calibration was applied to each sample measurement to calculate the molecular mass of each histogram distribution during analysis. WT apo Mm GS, Mm GS(R167L-A168G), and Bs GS, both in 25 mM Tris pH 7.5, 150 mM NaCl, 2.5% (v/v) glycerol, and 1 mM MgCl$_2$ were diluted in the working buffer in 20 mM Tris, 150 mM NaCl pH 7.5 to final concentrations of 150 nM prior to sample analyses with 2-fold dilution on buffer droplet to a final concentration of 75 nM. 10 μL of fresh buffer (20 mM Tris pH 7.5, 150 mM NaCl) adjusted to rt was pipetted into a well to find the focal position, which was identified and locked in using the autofocus function of the instrument. For each acquisition, 10 μL of diluted protein samples were added to the well and thoroughly mixed, and movies of 60 s duration with 6000 frames (10 frame rate/Hz) were recorded for each measurement using Refeyn AcquireMP v2023 R1 software using normal measurement mode with regular image acquisition settings. All mass photometry movies of each measurement were processed and analyzed by Refeyn DiscoverMP v2023 R2 software, and Gaussian curves were fit to each histogram distribution, and the mass (kDa), sigma (kDa), and normalized counts were determined. The average and standard deviation of molecular masses and its respective populations of four measurements of each group were calculated in excel (see source data file).

### Crystallization and structure determination of Mm GS(R167L-A168G)
Purified Mm GS(R167L-A168G) in buffer A was concentrated to 20 mg/mL and used for crystallization trials at rt with wizard screens I-IV, PEG Rx1, PEG Rx2 and Natrix screens. Crystals were produced by mixing the protein 1 to 1 (2 μL to 2 μL) with a crystallization condition consisting of 35% (v/v) PEG 200, 0.1 M sodium citrate, pH 5.5 and took a week to reach maximal size. The crystals could be cryo-preserved straight from the drop and data were collected at the Advanced Light Source (ALS) beamline 5.0.2 and processed with XDS[44] (Version January 10, 2022) (Supplementary Table 2). The structure was solved by molecular replacement (MR) using a one subunit of the apo Mm GS cryo-EM structure as a search model in Phenix (version 1.19)[45,46]. Three subunits were found in MR and crystallographic symmetry generates the GS dodecamer. The Arg167 and Ala168 side chains were truncated to alanines prior to the initial round of refinement. After the first 3 cycles

of refinement, the $R_{free}$ was 25%. After substituting the Arg167 and Ala168 side chains to leucine and glycine, respectively, and adding solvent, the structure was further refined to convergence. For data collection and refinement statistics see Supplementary Table 2.

## Crystallizations and structure determination of *Mm* GS·GlnK1

To form the complex prior to crystallization trials, purified *Mm* GS and *Mm* GlnK1 (in a buffer of 25 mM Tris pH 7.5, 150 mM NaCl, 5% (v/v) glycerol, 1 mM βME and 5 mM MgCl$_2$) were mixed ~2:1 (excess GlnK1 to GS) and loaded onto a S300 SEC column. The peak containing the complex (both proteins) (Supplementary Fig. 12) was obtained and concentrated to 35 mg/mL for crystallization trials. Crystals were obtained by mixing the GS-GlnK1 complex 1 to 1 with a crystallization reagent composed of 30% (v/v) PEG 400, 0.1 M sodium cacodylate pH 6.5 and 0.1 M LiSO$_4$. Crystals took from 3 days to several weeks to grow to maximum size and were cryopreserved from the drop. Data were collected at the Advanced Light Source (ALS) beamline 5.0.2 and processed with XDS[44] (Version January 10, 2022) (Supplementary Table 2). The structure was solved by molecular replacement (MR) using a hexamer of the *Bs* glutamine structure as a search model in Phenix (version 1.19)[45] PDB: 4LNN. Crystallographic symmetry generates a dodecamer. After an initial round of refinement in Phenix[45], and replacement of *Mm* side chains in Coot (version 0.9.6)[46], clear density was observed for residues in the GlnK1 T-loop. The GlnK1 trimer density was present but weak, which is consistent with the finding that the GlnK1 trimer body makes no contacts in the crystals (Supplementary Fig. 13). In addition, the crystals are twinned with two GlnK1 trimer body conformations in the crystal. Refinement with the twin operator (h,-k,-l) revealed in Xtriage led to improved electron density maps (Supplementary Fig. 14). Construction of the GlnK1 T-loop and GlnK1 trimer region reduced the $R_{free}$ and the structure was then subjected to additional refinement before ordered solvent was placed. Molprobity (version 4.5.1) was used to validate throughout refinement. See Supplementary Table 2 for data collection and refinement statistics.

## Cryo-EM sample and grid preparation of apo *Mm* GS

Purified *Mm* GS was buffer exchanged into Buffer B (12.5 mM Tris pH 7.5, 5 mM MgCl$_2$, 150 mM NaCl, 2.5% (v/v) glycerol, 1 mM βME) using a 10 kDa molecular weight cutoff (MWCO) spin filter (Millipore). For grid preparation, UltrAuFoil R1.2/1.3 Au 300 (Quantifoil) holey gold grids were cleaned for 300 s using a PELCO easiGlow glow discharge cleaning system and 3 μL of the GS sample (1.0 mg/mL) were applied at rt at 90% humidity. Following a 10 s incubation period, the grids were blotted for 1.5 s and plunged into liquid ethane using a Leica EM GP2 (Leica Microsystems). We noted that while the initial sample contained added sP26, either this peptide was dislodged during grid formation or did not bind effectively. At concentrations of 1 mg/mL in buffer with 1 mM MgCl$_2$, at least half the population appeared to be monomers or dimers too small to analyze. As increasing Mg concentrations aids in oligomer stabilization, we used 5 mM MgCl$_2$ to obtain these images.

## Cryo-EM data acquisition and single particle analysis for apo *Mm* GS

Partial and apo *Mm* GS: The cryo-EM dataset was collected on a 300 keV Titan Krios G3i (Duke) equipped with a Gatan K3 direct-electron detector. Automated data collection was performed[47]. A total of 6,624 movies were collected at a nominal magnification of 22,500X, giving a pixel size of 1.08 Å. The cryo-EM dataset was processed with cryoSPARC v3[48]. Movies were gain, motion, and CTF corrected. Initial 2D classes were generated using Blob Picker on a subset of 100 micrographs. Classes with well-centered particles of interest and visible elements of secondary structure were used to pick particles with the template picker and generate an Ab-initio map without symmetry applied. 2D classification was performed and EM maps were generated

by multiple rounds of homogenous refinement, global CTF Refinement, and local CTF refinement until no improvement in resolution was obtained. Heterogeneous refinement with 3 classes allowed for separate reconstructions of the apo GS and partial GS complexes to be obtained from the same dataset. The number of micrographs, number of particles, symmetry, resolution, and beta factor for each map is provided in Supplementary Table 1.

## Cryo-EM model building and refinement for the apo *Mm* GS

To model the *Mm* GS complexes, first, one GS subunit from the published *S. aureus* GS-glutamine-GlnR peptide structure, 7TF6[18] was docked in the maps using UCSF Chimera X 1.5[49]. GS residues were mutated to match the correct sequence using Coot[46]. Multiple rounds of fitting in Coot[46] and real-space refinement in Phenix[45] were performed to improve the quality of the models. After fitting one subunit, multiple copies of the GS were generated and docked into the maps to complete the dodecamer and partial oligomer complexes. This was followed by a final round of refinement in Phenix[45] and fitting in Coot[46].

## Cryo-EM sample and grid preparation of *Mm* GS-Met-Sox-P- ADP complex

Purified *Mm* GS was buffer exchanged into Buffer B (12.5 mM Tris pH 7.5, 5 mM MgCl$_2$, 150 mM NaCl, 2.5% (v/v) glycerol, 1 mM BME) using a 50 kDa MWCO spin filter (Millipore). To generate the GS-Met-Sox-P complex, *Mm* GS was concentrated to 10 mg/mL and mixed with ATP and L-methionine-S-sulfoximine (Met-Sox) to give final concentrations of 5 mM of both ATP and Met-Sox. The mixture was allowed to incubate at rt for 1 hr prior to grid preparation. For grid preparation, the reaction was diluted to 1 mg/mL with Buffer B and applied to UltrAuFoil R1.2/1.3 Au 300 (Quantifoil) holey gold grids. First the grids were cleaned for 300 s using a PELCO easiGlow glow discharge cleaning system and then 3 μL of the sample was applied at rt at 90% humidity. Following a 10 s incubation period, the grids were blotted for 1.5 s and plunged frozen into liquid ethane using a Leica EM GP2 (Leica Microsystems).

## Data collection and processing for Transition State of *Mm* GS-Met-Sox-P ADP complex

Data were screened and collected at the UNC Cryo-electron Microscopy core, using a 200 kV Talos Arctica. The scope had been modified to house a Gatan K3 5,760 ×4,092 pixel direct electron detector, which allowed for data to be collected at a pixel size of 0.87 Å or 45000X normal magnification[50]. 2764 movies were collected using SerialEM[47] software before being transferred to Duke University via Globus Connect to be analyzed using cryoSPARC v3[48]. Movies were gain, motion, and CTF corrected before the Manual picker function was used to select 107 particles for the creation of a template. Two templates resulted that represented the two primary views of the protein, the top/bottom view and the side view. The Template picker, Inspect Particle Picks and Extract micrograph functions were used to select 198,986 particles for 2D classification. This resulted in the creation of 30 2D classes being created. Seven of these classes, which represented 62% of the particles (123,421), where selected for further refinement based on the visibility and representation of the multiple viewpoints of the particle. Initial runs of Ab-Initio with no symmetry imposed followed by Homogeneous Refinement with D6 symmetry imposed resulted in the creation of a protein map with a resolution of 2.98 Å. This was improved to 2.7 Å by applying the global CTF Refinement, and local CTF refinement functions to the data and repeating the Homogenous Refinement with D6 symmetry.

## Cryo-EM model building and refinement for the *Mm* GS-Met-Sox-P ADP complex

To model the GS-Met-Sox-P ADP complex the high resolution *Pp* GS-Met-Sox-P ADP complex was docked into the density (as the density

revealed it had the same conformation as the *B. subtilis, Pp, L. mono-cytogenes* and *S. aureus* Met-Sox-P ADP complexes) using Coot[46]. The GS residues were then mutated to match the correct sequence in *Mm* GS in Coot[46]. Multiple rounds of fitting in Coot[46] and real-space refinement in Phenix[45] were performed to improve the quality of the model.

## Reporting summary

Further information on research design is available in the Nature Portfolio Reporting Summary linked to this article.

## Data availability

The structural data generated in this study by cryo-EM have been deposited in the Protein Data Bank under the codes 8TFC and 8TFB for the partial and dodecamer apo *Mm* GS structures and 8TFK for the *Mm* GS-Met-Sox-P-ADP cryo-EM structure. The coordinates and structure factor amplitudes for the *Mm* GS-GlnK1 and the *Mm* GS(R167L-A168G) crystal structures have been deposited in the Protein Data Bank under the accession codes 8TGE and 8UFJ, respectively. Additional publicly available PDB entries mentioned in this paper include: 7TF6 and 4LNN. The cryo-EM maps have been deposited in the Electron Microscopy Data Bank (EMDB) under accession codes EMD-41229, EMD-41228, and EMD-41232. Other source data are provided in the source data file. Source data are provided with this paper.

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

## Acknowledgements

This research was supported by National Institutes of Health grants (R35-GM130290 to M.A.S.). We acknowledge beamline 5.0.1 and 5.0.2 for X-ray diffraction data collection. The ALS (Berkeley, CA) is a national user facility operated by Lawrence Berkeley National Laboratory on behalf of the US Department of Energy under Contract DE-AC02-05CH11231, Office of Basic Energy Sciences. Beamline 5.0.2 of the ALS, a US Department of Energy Office of Science User Facility under Contract DE-AC02-05CH11231, is supported in part by the ALS-ENABLE program funded by the NIH, National Institute of General Medical Sciences, Grant P30 GM124169-01. We acknowledge Dr. Joshua Strauss of the UNC Cryo-EM Core Facility for technical assistance in this project. Data were collected at the UNC at Chapel Hill Cryo-EM Core Facility with a 200 kV Thermo Fisher Scientific Talos Arctica G3 equipped with a Gatan K3 direct electron detector. Mass photometry was performed in the UNC Macromolecular Crystallography core lab supported by the National Cancer Institute of the National Institutes of Health under award number P30CA016086.

## Author contributions

M.A.S. performed FP, solved the *Mm* GS-GlnK1 and *Mm* GS(R167L-A168G) crystal structures, constructed the *Mm* GS TS model, designed experiments, generated figures and wrote the paper. B.A.T. made samples for cryo-EM data analyses and solved apo *Mm* GS cryo-EM structures. R.S. purified proteins, performed SEC, MP and enzyme assays. R.R.S. carried out MP data collection with R.S. and performed MP analyses. N.L. purified proteins, generated the cryo-EM map for the *Mm* GS TS structure and aided in figure generation.

## Competing interests

The authors declare no competing interests.
