## [Peer Review File · Nature Communications]

Reviewers' Comments:

Reviewer #1:

Remarks to the Author:

This is a well-written, high-quality manuscript on the structure and regulation of an archaeal glutamine synthetase, a critical enzyme of nitrogen metabolism in all living organisms. The manuscript describes i) a novel, unstable intermediate of GlnA and ii) stabilization of this intermediate via the interaction with GlnK1, a PII-type regulatory protein. Thus, this work identifies a novel form of regulation of activity of glutamine synthetases and a novel function of widespread PII-type proteins.

I have only editing comments, some of them very minor.

1. The description of the fluorescence polarization assay is unclear. What binds to what, which binding is measured, and what is the readout?
2. PII proteins normally interact with ATP and 2-oxoglutarate. How is it relevant to the observed complex formation with GlnA?
3. Several manuscript figures show a gap in the tetrameric ring structure (e.g., Fig. 2A), but the cryo-EM image (Fig. 1B) indicates that all four subunits touch each other. Please, explain. If the gap is real, what is holding together the two dimers in the tetramer structure?
4. The authors' previous observation of a dodecamer to tetradecamer oligomeric transition in some bacterial GlnAs is, likely, relevant for this manuscript and should be mentioned.
5. The ability of P26 to form a complex with GlnA and GlnK1 should, likely, be mentioned.
6. "Archaeal" should be included in the title or abstract or both.
7. P. 3, lines 8-9: GS is not a gene.
8. P. 4, line 21 and earlier: is adenylation different from AMPylation?
9. P. 5, line 14: please, explain or rephrase "on each end".
10. P. 6, lines 8-9 and elsewhere: hexamer rings cannot have four subunits.
11. P. 8, line 5: what is "the transition state"?
12. P. 9, line 19: change "purified" to "created and purified".
13. P. 11, line 15 requires a reference.
14. P. 11, line 17: delete "in".
15. P. 11-12: what is the total similarity between GlnK1 and GlnK2?
16. Please, use continuous line numbering for manuscript submissions.

Reviewer #2:

Remarks to the Author:

In this manuscript, Schumacher and co-workers report cryo-EM structures of apo *Methanosarcina mazei* (Mm) glutamine synthetase (GS) and of the Mm GS transition complex with Met-Sox-P and ADP, as well as the crystallographic structure of the complex between Mm GS and the regulatory protein GlnK1. Through a combination of structural analysis and biochemical characterization, the authors propose a novel enzyme regulation mechanism based on the stabilization of high-order oligomeric species of Mm GS.

This work is relevant to the field of nitrogen metabolism and both the structural and biochemical studies convincingly support the conclusions. In addition, the manuscript is in general well written and data analysis seems to be well conducted. There are, however, some questions that the authors are requested to address:

- 1) In the introduction (page 3, line13), it is correctly mentioned that eukaryotic GSII enzymes possess a decameric organization but the work by Llorca and co-workers (ref. 11) in incorrectly

referenced in this context, as it reported a – never confirmed – octameric organization for *Phaseolus vulgaris* GSII.

2) In Fig. 1A there are “fibrils” or “tubes” that seem to be formed by GS oligomers. Can the authors please comment on this? Could this arrangement be relevant in vivo or have any implication for the function of the protein?

3) In page 9, line 13 the authors claim that the activity of the single amino acid variants of MmGS they engineered was “essentially abrogated”, when in fact it was ~5-fold lower than the wild type protein. Please rephrase.

4) In the Methods section, the MW range of the standards used for calculating the standard curve for the SEC experiments is inadequate. The experiment should be repeated with a suitable calibration curve.

5) Although the methods section details how the Mm GS-GlnK1 complex was formed, it would be interesting to show the analysis by SEC of the interaction between both proteins in an additional supplemental figure.

6) In the methods section, the authors state that in the crystal structure of Mm GS-GlnK1 complex, the GlnK1 trimer density is present but weak, which the authors attribute to two possible conformations of each GlnK1 trimer. This is in agreement with some of the statistics of the wwPDB validation report. The authors are requested to discuss this further, and to clarify whether the two conformations were modeled and refined. They should also update Table S2 to include the mean B-factors values for each chain in the structure.

7) In the wwPDB validation report for the crystallographic structure, although the L-test results seem OK, Xtriage suggests the presence of nearly perfect twinning (with law $h, -k, -l$). Given the reported space group, this would be unusual but not impossible, and the authors are requested to explain it. Importantly, could this be related to the dual-conformation of the GlnK1 trimer (see point 6, above)?

8) The supplemental Fig. S2 provided is identical to Fig. 1; please provide the correct figure.

9) The authors are urged to deposit their raw images (both cryoEM and X-ray diffraction) in public repositories (e.g., EMPIAR, SBGrid Data Bank, IRRMC).

Minor comments:

1) Please revise the sentence in page 6, lines 19-21.

2) Please include labels in Fig. 3 to better distinguish both proteins (similar to Fig. 4).

3) It would be useful to indicate in the overall molecule where the region depicted in Fig. 4A is located (maybe zooming into this region on an adjacent representation of the oligomer). Further, the figure would be much more informative if presented as a stereogram.

4) The interactions of residue Gln199 are described in the text (page 9, line 7) but not represented in the corresponding figure (Fig. 4A, not 4B as indicated in the text).

5) Figure 6D is mis-referenced in page 10, line 8; please revise.

6) Please revise the sentence in page 12, line 1.

7) “Cs”, presumably for *Camellia sinensis*, is used in page 14, line 9 but is not previously defined.

8) In the Methods section, please provide the manufacturer for the protease inhibitor cocktail

used.

9) In the Methods section, please provide the manufacturer and size for the cobalt NTA column used.

10) In the Methods section, please specify the volume of lysate (and the originating volume of bacterial culture) loaded in the IMAC column.

11) In the Methods section, please specify the nature (molar? mass?) of the GS:GlnK1 ratio and, if molar, specify if of which species (monomer, multimer?).

12) In sP26 is mentioned in the Methods section (page 18, line 22), but not elsewhere in the text; please revise.

13) In the Methods section, the references for several programs are not consistently indicated when those programs are mentioned; please revise.

14) Please revise the sentence in page 20, lines 11-13.

15) In the legend of Fig. S1, please indicate that the loop regions are colored blue and indicate the meaning of the residues colored yellow. Also, a region "boxed in red" is mentioned in the legend but not displayed in the figure. Finally, there is a single reference to Mm GS as "Mm GlnA1 protein (Mm GS)", which is not used anywhere else in the text.

16) Please, correct the following typos:

- Please italicize "Mm" in line 19 of page 5, in Fig. S1 and its legend, and in line 11 of page 29.
- "Figure 1A-B" in line 17 of page 6 should be "Figure 2A-B".
- "Ser251" in line 1 of page 9 should be "Ser252".
- "Figure 4B" in line 8 of page 9 should be "Figure 4A".
- "SPERDEX" in line 16 of page 15 should be "SUPERDEX".
- "AKTA prime plus" in line 16 of page 15 should be "ÄKTAprime plus".
- "37 °C" in line 15 of page 16 should read "37°C".
- In the Methods section, please define "MWCO" and "rt".
- Is it "cryoSPARC v3" (page 19, line 10) or "Cryosparc V3" (page 21, line 3)?
- On page 21, line 8, please replace "7" with "Seven".
- "uM" in the x-axis label of Fig. 6D should be "µM".
- Figure S1 should not be referenced in line 4 of page 6.
- The Y179 loop is defined by residues 154-161 (line 18 on page 6) but in Fig. S1 seems to be defined by residues 152-161. Please check.
- Please replace B3 by β3 in Figure 6C.

Reviewer #3:

Remarks to the Author:

Below we summarize revisions and address the critiques from the reviewers (our comments in
red). We would like to thank all the referees for their excellent comments and suggestions.

4 **REVIEWER COMMENTS**

**Reviewer #1 (Remarks to the Author):**

This is a well-written, high-quality manuscript on the structure and regulation of an archaeal
glutamine synthetase, a critical enzyme of nitrogen metabolism in all living organisms. The
manuscript describes i) a novel, unstable intermediate of GlnA and ii) stabilization of this
intermediate via the interaction with GlnK1, a PII-type regulatory protein. Thus, this work
identifies a novel form of regulation of activity of glutamine synthetases and a novel function of
widespread PII-type proteins.

I have only editing comments, some of them very minor.

**We appreciate the referee's positive comments and very helpful suggestions.**

1. The description of the fluorescence polarization assay is unclear. What binds to what, which
binding is measured, and what is the readout?

**We have included more details in the Methods and in the text regarding this assay. In a
fluorescence polarization (FP) assay, a small fluorescently labeled molecule (here ATP) that is
excited with plane-polarized light emits mostly depolarized light due to its rapid tumbling. Upon
binding a larger target protein (GS), the F-molecule's rotation is slowed, and the emitted light
becomes polarized. Binding saturation is indicated by plateau of signal. Addition of a second
protein (that does not bind the F-molecule, here GlnK1) but binds the protein, further slows the
molecule rotation until a second binding saturation is reached, as indicated by another plateau of
the change in polarization. As noted, we have added more description of this in the text and
Methods.**

2. PII proteins normally interact with ATP and 2-oxoglutarate. How is it relevant to the observed
complex formation with GlnA?

**ATP and oxoglutarate are not needed to form the complex, this information is now included.**

3. Several manuscript figures show a gap in the tetrameric ring structure (e.g., Fig. 2A), but the
cryo-EM image (Fig. 1B) indicates that all four subunits touch each other. Please, explain. If the
gap is real, what is holding together the two dimers in the tetramer structure? **The subunits all
make interactions in the tetramer of dimers (four subunits in each layer) that hold together the
tetramer on top but also make interactions with the subunits below via the thong interactions.
Because each top oligomer, which forms a hexamer in the active enzyme, is missing two subunits
in this form, there is an opening at the location of where these subunits normally dock in the
dodecamer with two hexamers. But the only exposed part of this partial oligomer is at the
subunits that have one interaction with another subunit. However, again these are held in place
by tight contacts on one side in the hexamer ring as well as the thong interactions they make with
the subunits on the layer below. There are no subunits that are completely exposed and not
involved in contacts with neighboring subunits.**

**To further probe the formation of oligomers of *Mm* GS in the apo form, we performed new Mass
photometry (MP) experiments that we now include. MP utilizes low concentrations of sample that
matches the physiological concentrations of many proteins. The method provides molecular
weight information and thus data on oligomeric states (the method cannot be used for analyzing
protein-protein complexes that form with K_d s in the μ M range because of its requirement for low
concentrations). We obtained MP data for the WT apo *Mm* GS and compared that to the Gram-**

positive apo *Bacillus subtilis* GS. Strikingly, these experiments show that, at 75 nM, the apo *Mm* GS
forms essentially no dodecamers while the *B. subtilis* GS sample showed >60% dodecamer, which
supports our data.

4. The authors' previous observation of a dodecamer to tetradecamer oligomeric transition in
some bacterial GlnAs is, likely, relevant for this manuscript and should be mentioned.
We thank the reviewer for this comment. This information has been added to the discussion.

5. The ability of P26 to form a complex with GlnA and GlnK1 should, likely, be mentioned.
This has been added to the introduction.

6. "Archaeal" should be included in the title or abstract or both.
We have included this in the abstract and also (*M. mazei*) in the title. We note that we also
changed the title as it was too long (it is now 15 words).

7. P. 3, lines 8-9: GS is not a gene.
The reviewer is correct, we have altered this sentence. Thanks for pointing out this typo.

8. P. 4, line 21 and earlier: is adenylylation different from AMPylation?
These are the same terms for this modification, we have tried to clarify this.

9. P. 5, line 14: please, explain or rephrase "on each end".
We have replaced with "on each hexamer face". We thank the reviewer for noting the need for
clarification.

10. P. 6, lines 8-9 and elsewhere: hexamer rings cannot have four subunits.
We have rephrased to indicate in the revision that there are four subunits within each ring of the
partial oligomer.

11. P. 8, line 5: what is "the transition state"?
Methionine sulfoximine (MSO) can be phosphorylated by GS to form what has been shown to be a
transition state analog. This analogue traps the enzyme in this state as it is unable to diffuse from
the enzyme active site. In other words, this Met-Sox-P complex mimics the gamma-glutamyl
phosphate transition state. This was reported in structural detail first by Krajewski et al., Structure
of a *Mycobacterium tuberculosis* glutamine synthetase in complex with a transition-state mimic
provides functional insights. Proc. Natl. Acad. Sci. USA, 102: 10499-10504 (2005).

12. P. 9, line 19: change "purified" to "created and purified".
done

13. P. 11, line 15 requires a reference.
Added.

14. P. 11, line 17: delete "in".
Done

15. P. 11-12: what is the total similarity between GlnK1 and GlnK2?
We should have included this, and we thank the reviewer for this comment. We have now added
this information in the text.

16. Please, use continuous line numbering for manuscript submissions.

**We apologize for this and have now made sure the document contains continuous line numbers.**
**Again, we wish to thank this reviewer for their very helpful comments.**

**Reviewer #2 (Remarks to the Author):**

In this manuscript, Schumacher and co-workers report cryo-EM structures of apo *Methanosarcina*
*mazei* (Mm) glutamine synthetase (GS) and of the Mm GS transition complex with Met-Sox-P and
ADP, as well as the crystallographic structure of the complex between Mm GS and the regulatory
protein GlnK1. Through a combination of structural analysis and biochemical characterization, the
authors propose a novel enzyme regulation mechanism based on the stabilization of high-order
oligomeric species of Mm GS. This work is relevant to the field of nitrogen metabolism and both
the structural and biochemical studies convincingly support the conclusions. In addition, the
manuscript is in general well written and data analysis seems to be well conducted. There are,
however, some questions that the authors are requested to address:

**We thank the reviewer for their excellent comments and suggestions.**

1) In the introduction (page 3, line13), it is correctly mentioned that eukaryotic GSII enzymes
possess a decameric organization but the work by Llorca and co-workers (ref. 11) in incorrectly
referenced in this context, as it reported a – never confirmed – octameric organization
for *Phaseolus vulgaris* GSII.

**Thanks to the reviewer for this point. We have clarified this in the revision.**

2) In Fig. 1A there are “fibrils” or “tubes” that seem to be formed by GS oligomers. Can the authors
please comment on this? Could this arrangement be relevant in vivo or have any implication for
the function of the protein?

**This is an interesting question. There have been reports that *E. coli* GS can form tubes in the**
**presence of metals (Dabrowski et al., *Biochemistry* **33**, 14957-14964 (1994)) and the packing of the**
***S. cerevisiae* GS crystal structure showed extended fibrils. However, there is presently, to our**
**knowledge, no evidence for any physiological role for such tubes. We note, that the *Mm* GS does**
**not harbor the metal binding residues proposed to mediate tube formation in the *E. coli* GS and**
**metals like zinc or copper are not involved in the apparent formation of the *Mm* GS tube. The**
**group working on the *S. cerevisiae* GS, who published their fibril crystal observation in 2009,**
**indicated that they were looking into physiological roles, but have not since released any follow up**
**data on possible physiological roles. We note that we do not see evidence for the formation of**
**such tubes of either *Mm* GS at low concentrations using mass photometry (MP) (see point 4)**
**below) so we do not wish to speculate at this point on any possible function as these tubes may be**
**artifactual (formed at higher concentrations).**

3) In page 9, line 13 the authors claim that the activity of the single amino acid variants of MmGS
they engineered was “essentially abrogated”, when in fact it was ~5-fold lower than the wild type
protein. Please rephrase.

**This has been rephrased as requested.**

4) In the Methods section, the MW range of the standards used for calculating the standard curve
for the SEC experiments is inadequate. The experiment should be repeated with a suitable
calibration curve.

**We used 6 standards to generate the calibration curve, now included. To further probe the**
**formation of oligomers of *Mm* GS, we performed new Mass photometry (MP) experiments that**

we now include. MP utilizes low concentrations of sample that matches the physiological
concentrations of many proteins. The method provides molecular weight information and thus
data on oligomeric states (the method cannot be used for analyzing protein-protein complexes
that form with K_d s in the μ M range because of its requirement for low concentrations). We
obtained MP data for the WT apo *Mm* GS and compared that to the Gram-positive apo *Bacillus*
*subtilis* GS. Strikingly, these experiments show that, at 75 nM, the apo *Mm* GS forms essentially no
dodecamers while the *B. subtilis* GS sample showed >60% dodecamer, which supports our data.

5) Although the methods section details how the *Mm* GS-GlnK1 complex was formed, it would be
interesting to show the analysis by SEC of the interaction between both proteins in an additional
supplemental figure.

This has been included. For this experiment, since we did not know the stoichiometry, we first
purified both proteins, mixed them together with approx. 2 fold excess GlnK1 (to make sure the
GS was saturated with GlnK1) then ran them on a preparative SEC column. We did not perform
162 MW analyses and used the sample for crystallization trials. The GS-GlnK1 complex eluted
essentially at a molar ratio of 2 to 1 GS (monomer) to GlnK1 (monomer) despite starting with
excess GlnK1. This was observed in the structure where there are two GlnK1 trimers to a GS
dodecamer. We note also that when we just mixed the samples of GS and GlnK1 approx 1:1 and
concentrated the complex, these samples also produced the identical crystals as we obtain with
protein from SEC.

6) In the methods section, the authors state that in the crystal structure of *Mm* GS-GlnK1 complex,
the GlnK1 trimer density is present but weak, which the authors attribute to two possible
conformations of each GlnK1 trimer. This is in agreement with some of the statistics of the wwPDB
validation report. The authors are requested to discuss this further, and to clarify whether the two
conformations were modeled and refined. They should also update Table S2 to include the mean
B-factors values for each chain in the structure.

We have added the mean B-values for all the chains in Table S2. For the GlnK1 subunits we also
included the B-ave for the T-loops that are bound to GS. We have discussed the structure
determination of the GS-GlnK1 complex in more detail (see response to point 7) below.

7) In the wwPDB validation report for the crystallographic structure, although the L-test results
seem OK, Xtriage suggests the presence of nearly perfect twinning (with law $h, -k, -l$). Given the
reported space group, this would be unusual but not impossible, and the authors are requested to
explain it. Importantly, could this be related to the dual-conformation of the GlnK1 trimer (see
point 6, above)?

We thank the reviewer for this point. We re-refined the structure with the twin law relating the
two GlnK1 conformations; this resulted in improved density for GlnK1. We replaced the original
coordinates in the pdb with these coordinates. While the overall structure remained essentially
the same, we used the new coordinates to remake all figures for the GS-GlnK1 complex. We have
also added two supplemental figures to show 1) the packing, which shows that the GlnK1 trimers
do not make any crystal contacts – are held in place by interactions with their partner GS, and 2)
SigmaA weighted (less biased) 2mFo-DFc maps for the GlnK1 trimer regions and T-loops.

8) The supplemental Fig. S2 provided is identical to Fig. 1; please provide the correct figure.

We thank the reviewer very much for catching this. Supplemental Figure 2 was supposed to be the
cryo-EM data processing and reconstruction details for the dodecamer form of apo GS. The
correct Figure has now been included.

9) The authors are urged to deposit their raw images (both cryoEM and X-ray diffraction) in public
repositories (e.g., EMPIAR, SBGrid Data Bank, IRRMC).

Thank the reviewer for the suggestion. We are looking into data image storage. In particular, this
will be helpful for cryo-EM data as storage is not provided by Duke.

Minor comments:

1) Please revise the sentence in page 6, lines 19-21.

Done

2) Please include labels in Fig. 3 to better distinguish both proteins (similar to Fig. 4).

This was a little trickier than Fig. 4 as we show slightly different colors for each interacting pair (i.e.
pink and magenta for the *S. aureus* GS and light_green and green for Mm GS). But we have
included these labels and they do help distinguish the proteins.

3) It would be useful to indicate in the overall molecule where the region depicted in Fig. 4A is
located (maybe zooming into this region on an adjacent representation of the oligomer). Further,
the figure would be much more informative if presented as a stereogram.

It is a good recommendation to show the location of the active site in the overall dodecamer with
a blow up of an active site. We thank the reviewer for the suggestion. We have now replaced Fig
4A with such a figure. Making this figure a stereogram, however, resulted in the figure being too
small to be legible.

4) The interactions of residue Gln199 are described in the text (page 9, line 7) but not represented
in the corresponding figure (Fig. 4A, not 4B as indicated in the text).

Residue 199 is a glutamic acid, Glu199. This is included in what should be called out as Figure 4A.

5) Figure 6D is mis-referenced in page 10, line 8; please revise.

We have fixed this. We thank the reviewer for catching this.

6) Please revise the sentence in page 12, line 1.

We have rewritten this sentence.

7) "Cs", presumably for *Camellia sinensis*, is used in page 14, line 9 but is not previously defined.

We have now included the full genus/species *Camellia sinensis* in the first instance of its usage.

8) In the Methods section, please provide the manufacturer for the protease inhibitor cocktail
used.

This information (Roche) and more detail on the cocktail has been added.

9) In the Methods section, please provide the manufacturer and size for the cobalt NTA column
used.

We have included this information in the revision.

10) In the Methods section, please specify the volume of lysate (and the originating volume of
bacterial culture) loaded in the IMAC column.

This information has been added.

11) In the Methods section, please specify the nature (molar? mass?) of the GS:GlnK1 ratio and, if
molar, specify if of which species (monomer, multimer?).

This information has been included in the methods. Also see response to 5) above.
12) In sP26 is mentioned in the Methods section (page 18, line 22), but not elsewhere in the text;
please revise.
We did not see tight binding of sP26 to GS and it was not our focus. However, we have added
some information in the revision about sP26, as requested.
13) In the Methods section, the references for several programs are not consistently indicated
when those programs are mentioned; please revise.
We apologize for this oversight and have included reference citations for all mentions of the
proteins.
14) Please revise the sentence in page 20, lines 11-13.
This sentence has been rewritten.
15) In the legend of Fig. S1, please indicate that the loop regions are colored blue and indicate the
meaning of the residues colored yellow. Also, a region “boxed in red” is mentioned in the legend
but not displayed in the figure. Finally, there is a single reference to Mm GS as “Mm GlnA1 protein
(Mm GS)”, which is not used anywhere else in the text. We thank the reviewer for pointing this
out. We have indicated the blue regions correspond to active site loop regions, regions involved in
GlnK1 binding are yellow and the black boxed region contains the residues showing structural
differences between the Mm GS and Gram-positive GS. We simply use “Mm GS” as in the rest of
the text to describe the Mm enzyme.
16) Please, correct the following typos:
- Please italicize “Mm” in line 19 of page 5, in Fig. S1 and its legend, and in line 11 of page 29.
Done.
- “Figure 1A-B” in line 17 of page 6 should be “Figure 2A-B”.
Thanks to the reviewer for catching this typo, which has been fixed.
- “Ser251” in line 1 of page 9 should be “Ser252”.
Fixed.
- “Figure 4B” in line 8 of page 9 should be “Figure 4A”.
Fixed.
- “SPERDEX” in line 16 of page 15 should be “SUPERDEX”.
Fixed.
- “AKTA prime plus” in line 16 of page 15 should be “ÄKTAprime plus”.
We have changed this as suggested.
- “37 °C” in line 15 of page 16 should read “37°C”.
Done.
- In the Methods section, please define “MWCO” and “rt”.
These have now been defined.
- Is it “cryoSPARC v3” (page 19, line 10) or “Cryosparc V3” (page 21, line 3)?
It is cryoSPARC v3, which has been utilized now in all instances.
- On page 21, line 8, please replace “7” with “Seven”.
Done.
- “uM” in the x-axis label of Fig. 6D should be “µM”.
This has been fixed.
- Figure S1 should not be referenced in line 4 of page 6.

This has been removed as suggested. Thanks to the reviewer for catching this.
- The Y179 loop is defined by residues 154-161 (line 18 on page 6) but in Fig. S1 seems to be
defined by residues 152-161. Please check.

This has been fixed.

- Please replace B3 by β 3 in Figure 6C.

This has been fixed.

Again, we would like to thank the reviewer for their helpful input.

Reviewer #3 (Remarks to the Author):

I co-reviewed this manuscript with one of the reviewers who provided the listed reports. This is
part of the Nature Communications initiative to facilitate training in peer review and to provide
appropriate recognition for Early Career Researchers who co-review manuscripts.

We appreciate the time spent by both the mentor and Early Career Researcher. We are very
pleased to be involved in this early career review experience.
